# Functionalized Carbon-Based Electrochemical Sensors for Food and Alcoholic Beverage Safety

**Zhongjie Yang** [1,2]**, Xiaofei Zhang** [3,]*** and Jun Guo** [1,4,]***

1   Wenzhou Key Laboratory of Biomaterials and Engineering, Wenzhou Institute, University of Chinese Academy of Sciences, Wenzhou 325000, China
2   School of Chemistry and Materials, Guizhou Normal University, Guiyang 550001, China
3   Institute of Materials Research and Engineering A*STAR 2 Fusionopolis Way, Innovis, #08-08, Singapore 138634, Singapore
4   State Key Laboratory of Separation Membranes and Membrane Processes, School of Chemistry, Tiangong University, Tianjin 300387, China
*   Correspondence: stone623717@outlook.com (X.Z.); junguo@tiangong.edu.cn (J.G.)

**Abstract:** Food is a necessity in people's lives. Equally importantly, alcoholic beverages are also highly demanded globally due to the indispensable role they play in cultural, social, and ritual events. However, the production of food and alcoholic beverages suffers from a variety of contaminants, such as toxins, pesticides, antibiotic residues, and heavy metals, which are seriously harmful to human beings. These urgent threats have raised the awareness of the need to improve product quality and safety via developing effective, rapid, and economical monitoring and detecting methods. Fortunately, due to their numerous advantages, including high sensitivity, short response time, low cost, and easy portability, electrochemistry sensors have made huge contributions to ensuring the quality of food and alcoholic beverages. The purpose of this review is to introduce applications of electrochemical sensors to foods and alcoholic beverages, and to highlight the important role of carbon-based materials (i.e., carbon dots, carbon nanotubes, and graphene) as electrochemical sensors in detecting various contaminants. In addition, the preparation methods of these carbon-based electrochemical sensors and corresponding detection mechanisms are discussed in detail. It is hoped that this review can inspire more innovative detection technologies for ensuring the safety of food and alcoholic beverages.

**Keywords:** carbon-based materials; electrochemical sensor; food safety; alcoholic beverage safety; chemical contaminants

## 1. Introduction

Food is a necessity for human beings and the ongoing improvement in living standards leads to increasingly higher requirements for food production. Consequently, food safety is regarded as a major issue involved in social progress, economic development, and public health [1,2]. Currently, food safety is mainly challenged by the excessive presence of illegal additives, pesticide residues, organic contamination, biological toxins, veterinary drugs, and heavy metals added during the preparation, post-treatment, and storage processes. Similarly, alcoholic beverages are a globally popular drink that contain different constituents, including ethanol, water, peptides, organic acids, and sugars. Possible contaminants, such as carcinogens, methanol, mycotoxin, and amine, pose great threats to human health, thus leading to an increase in the public awareness about the safety of food and alcoholic beverages. Therefore, there is an urgent need to develop advanced analytical techniques characterized by high sensitivity, short response time, low cost, and easy portability [3–6].

In recent decades, numerous analytical methods and equipment have been exploited to meet the increasing need for qualitative or/and quantitative detection of various chemical

contaminants and hazardous substances existing in foods and alcoholic beverages. The classical analytical methods can be mainly categorized as chromatography and spectroscopy disciplines. For instance, the commonly used chromatographic techniques include hydrodynamic chromatography (HDC), high-performance liquid chromatography (HPLC), size exclusion chromatography (SEC), gas chromatography (GC), HPLC-mass spectrometry-mass spectrometry (HPLC-MSMS), matrix-assisted laser desorption/ionization (MALDI-TOF), and gas chromatographic-MSMS (GC-MSMS) [7–10]. Although these chromatographic methods have advantages in terms of high separation efficiency, accurate quantitation ability, and excellent detection sensitivity, the extremely high cost of equipment and maintenance, and the complicated operation process involved, have seriously limited their ability to be applied to food and alcoholic beverages in a portable and fast way. Compared with chromatographic techniques, spectroscopic methods such as near-infrared (NIR), ultraviolet-visible (UV-vis), Raman, and fluorescence spectroscopies are more efficiently and conveniently operated and analyzed, even without the destruction of analytes. However, the disadvantages are also evident, such as the requirement of multistage pretreatment and the lack of wide applicability. Because of these shortcomings, novel advanced analysis methods are expected to realize the rapid and efficient analysis of contaminants and residues [11,12].

Benefiting from numerous advantages, such as rapid response, high selectivity, flexibility, and portability, electrochemical sensors have become powerful candidates via transducing the specific interaction between analytes and sensors into recognizable electrical signals. Moreover, electrochemical sensors are cheap, simple, and applicable for various types of analytes. Generally speaking, electrochemical sensors are composed of sensitive components, conversion components, electronic circuits, and structural accessories [13]. As one of the important components, the active electrode, which is typically composed of nanomaterials with multiple properties, determines the sensitivity and functionality of electrochemical sensors [14–16]. Among various types of electrodes, the advent of carbon nanomaterials-based electrodes involving carbon dots (CDs), carbon nanotubes (CNTs), and graphene have created new possibilities for fast, reliable, and economical detection of contaminants and toxins in foods and alcoholic beverages [17–19]. To ensure superior performances, additional ions, molecules, or polymers are usually added for functionalizing the raw carbon-based materials with enhanced electrocatalytic activity, electro-chemiluminescence, photoelectricity, etc. [20].

The currently reported literature has demonstrated the superiority of carbon-based electrochemical sensors in the application of food and alcohol detection [21,22], and some recently published reviews have also summarized the syntheses of carbon nanomaterial-based electrodes and their application performances [23–26]. However, very few reviews have systematically illustrated their design principle and working mechanism [27]. A deep understanding of the relationship between the structure of carbon-based materials and their detection performances, and their application scopes, is still lacking. Therefore, a timely overview of the above issues is highly desirable, and can offer useful guidance for developing new types of highly efficient electrochemical sensors. In this review, we firstly introduce the structures and characteristics of carbon material-based electrochemical sensors. Then, their working mechanisms are also discussed by presenting some prototype examples. Finally, the perspectives and challenges in this field are provided based on our personal understandings.

## 2. Electrochemical Techniques

Electrochemical sensors have become a widespread analysis technique with the merits of low cost, high accuracy, fast response, and good reliability, and have drawn substantial attention for the qualitative or/and quantitative analysis of foods and alcoholic beverages [28,29]. The design of high-quality electrochemical sensors needs a full understanding of their basic detection principles. Figure 1a shows that samples without pretreatment can be first added into the electrochemical cell. Then, upon applying electricity, the electrochem-

ical sensor converts the physical and chemical response into recognizable electrochemical response signals. Through analyzing the correlation between the detected substances and corresponding electronic signals, the contaminants in foods and alcoholic beverages can be quickly identified. As shown in Figure 1b, a typical three-electrode electrochemical system includes a working electrode (WE), a reference electrode (RE) (e.g., Ag/AgCl), and a counter electrode (CE) (e.g., carbon rod, platinum). Recently, functional carbon-based nanomaterials such as CDs, CNTs, graphene oxide (GO), and porous carbon were selected as candidates for the working electrode (WE) due to their excellent conductivity and durability [30,31]. Of note, the selectivity, linearity, sensitivity, stability, detection limit, dynamic range, and response time of the resultant electrochemical sensors can be dramatically improved by further regulating the composition and the structure of adopted carbon nanomaterials [32–34]. Additionally, versatile electrochemical analysis methods are also required to be established for efficiently extracting the unique information from different analytes in food and alcoholic beverages. Specifically, the current-type (generating an amperometric current), potential-type (generating a potentiometric potential), conductivity-type (measurably altering the conductive properties of a medium between electrodes), and impedimetric-type (measuring impedance through an electrochemical impedance spectroscopy method) are the four types of golden protocols [35–38].

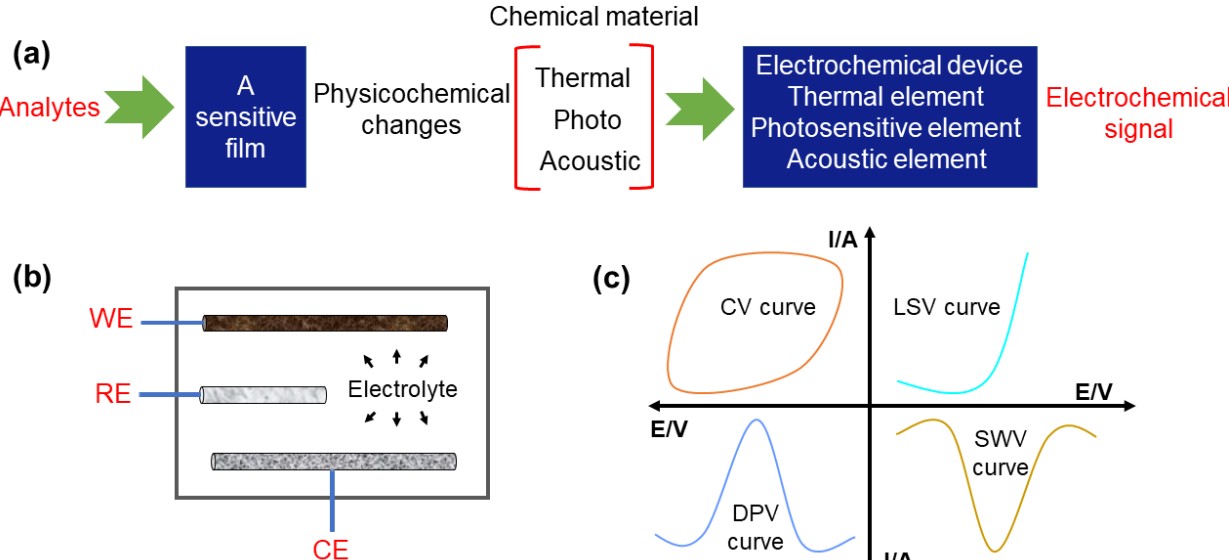

**Figure 1.** Principle and test methods of electrochemical sensors. (**a**) The detection principle of an electrochemical sensor. (**b**) The basic configuration of an electrochemical sensor. (**c**) The diagrams of cyclic voltammetry (CV), linear sweep voltammetry (LSV), differential pulse voltammetry (DPV), and square wave voltammetry (SWV) curves.

In practice, cyclic voltammetry (CV), linear sweep voltammetry (LSV), differential pulse voltammetry (DPV), and square wave voltammetry (SWV) (Figure 1c) are most commonly selected according to the properties of the detected substance [39–42]. So, it is necessary to understand the working principles of those electrochemical analysis methods, which is helpful for more rationally and accurately analyzing the contaminants in food and alcohol beverages. First, CV is a commonly used electrochemical analysis method operated by scanning the response current one or more times in a triangular waveform under varyingly applied voltage. The reversibility of electrode reaction, intermediate, phase boundary adsorption, phase transformation, and the mechanism of coupled reactions can be acquired and judged according to the presented shape of CV curves [43]. Second, LSV is another electrochemical analysis method in which a linear change in voltage is applied on the electrode; that is, the potential changes linearly with the applied voltage recording the current on the working electrode at the same time. The peak current measured in

the LSV curve has a linear relationship with the concentration of the measured analytes, which is the basis of quantitative analysis [44]. Third, DPV is an electrochemical method that involves the superposition of a linearly increasing voltage with a rectangular pulse of constant amplitude. As a result of the rapid development of electronic circuits, pulse voltammetry has been widely used in the field of electrochemical analysis. For instance, the quantitative determination of various analytes and the mechanism of complex electrode reactions can be decoded due to pulse voltammetry's high sensitivity and low detection limit. Moreover, by combining differential pulse voltammetry with other methods such as stripping voltammetry, the detection sensitivity can be greatly improved [45]. Finally, SWV is an effective electrochemical technique that can be utilized for electrode mechanism understanding and electro-kinetics measurement. It is worth mentioning that SWV can efficiently suppress the background current, thereby obviously increasing the signal-to-noise ratio and reducing the detection limit [46,47].

## 3. Carbon Nanomaterial-Based Electrochemical Sensors

Carbon is one of the most abundant elements in nature, and carbon-based materials (Figure 2) have been continuously studied and utilized in different fields. Generally speaking, according to the dimension of the carbon-based materials, they can be divided into zero-dimensional materials (their dimensions in all directions are in the order of nanometers, e.g., CDs), one-dimensional materials (their dimensions in two directions are in the order of nanometers, e.g., CNTs and carbon nanofibers), and two-dimensional materials (their dimension in only one direction is in the order of nanometers, e.g., graphene and GO) [48–55]. As early as the 18th century, it was known that graphite was formed by $sp^2$ and diamond was formed by $sp^3$ hybridized carbon atoms. Furthermore, the fullerenes, such as $C_{60}$ and $C_{70}$, and CNTs, were discovered in 1985 and 1991, respectively [56,57]. These studies not only expanded the scope of the carbon material family, but also marked the start of a new era for the research of carbon nanomaterials. In particular, the discovery of graphene in 2004 triggered a new wave of research [58]. Due to their excellent electrical conductivity, wide potential window, high electrocatalytic activity, and chemical modifiability, carbon-based materials have been accepted as one of the best candidates for the construction of electrochemical sensors [59]. In the following section, we summarize the exploration and application of typical carbon-based materials, including CDs, CNTs, and graphene, as electrochemical sensors in food and beverage safety due to their low cost, good conductivity, and facile fabrication process.

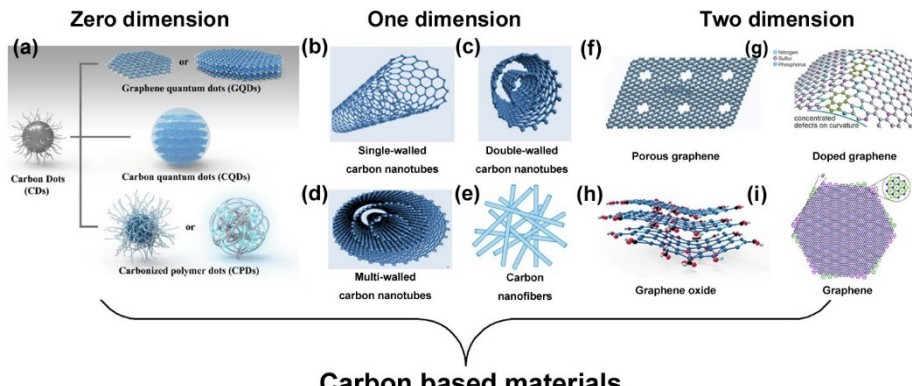

**Figure 2.** Classification of carbon-based sensors according to the dimension of materials. (**a**) Reproduced with permission from [48]. Copyright American Chemical Society, 2020. (**b**–**d**) Reproduced with permission from [49]. Copyright Springer Nature, 2019. (**e**) Reproduced with permission from [50]. Copyright MDPI, 2019. (**f**) Reproduced with permission from [51]. Copyright Elsevier, 2012. (**g**) Reproduced with permission from [52]. Copyright Wiley, 2016. (**h**) Reproduced with permission from [53]. Copyright Elsevier, 20. (**i**) Reproduced with permission from [54]. Copyright Springer Nature, 2020.

*3.1. CDs*

In 2004, Xu et al. discovered that CDs can present fluorescence during the process of electrophoresis [56]. CDs are defined as zero-dimensional nanoparticles having a size of less than 10 nm that are mainly composed of carbon elements. CDs have been widely used in electroluminescence, medical imaging, environmental monitoring, chemical analysis, photocatalysis, and energy conversion due to their colorful optical properties, good water solubility, low toxicity, environmental friendliness, abundant raw source, low cost, and good biocompatibility [60,61]. Regarding CDs' preparation, a variety of methods, including arc discharge, laser ablation, electrochemical oxidation, acidic oxidation, microwave, ultrasonic, calcination, hydrothermal, template synthesis, and hydrosol condensation polymerization, have been reported [62–64]. Interestingly, CDs can be functionalized easily with other materials, such as organic, polymeric, and inorganic species, via reacting with surface carboxylic acid moieties. In the following section, we illustrate the applications of CD-based materials in the development of electrochemical sensors for $H_2O_2$, dopamine (DA), triclosan, glucose, acetaminophen, $Cu^{2+}$ ions, uric acid, etc.

Acting as a neuromodulator of ionotropic synapses, DA sets a threshold for the striatal activity involved in many diseases and drug addiction. In addition, DA is available as an intravenous medication acting on the sympathetic nervous system. Hence, the determination of DA in vivo/vitro becomes increasingly important in practice [65,66]. As shown in Figure 3a, Zhu et al. [67] prepared a new type of N-doped CDs by a one-step microwave irradiation method. The N-doped CDs obtained by this method had a highly sensitive electrochemical response to DA, with a linear range of 0.05~8 μM and a detection limit of 1.2 nM. Moreover, in contrast to traditional detection methods requiring complex pretreatment and producing a large quantity of organic waste, the developed electrochemical sensing detection can be performed by directly adding DA to the detection system for qualitative and quantitative analysis. Huang et al. [68] designed a new type of DA sensor (Figure 3b) based on a Au@CDs-CS/glassy carbon electrode (GCE), which has high sensitivity and excellent performance stability and can suppress the background interference currents from ascorbic acid (AA) and uric acid (UA). Under the optimum experimental conditions, the linear range of 0.01~100.0 μM and the detection limit of 0.1 nM (S/N = 3) were obtained. Dai and co-workers [69] reported that CDs and chitosan were combined to construct a composite electrode. It was found that the signal was significantly higher than that of the bare electrode. Moreover, the composite electrode has higher sensitivity for the detection of triclosan, and also shows good results for the detection of actual samples such as toothpaste and gargle daily water, with the linear range of 10~1.0 mM and the detection limit of 0.92 nM (Figure 3c). As shown in Figure 3d, Sheng et al. [70] used CDs-Chitosan (CS)/hemoglobin composite membrane for electrochemical detection of $H_2O_2$, which showed good sensitivity and robust usability. The linear current response for $H_2O_2$ was from 1 to 118 μM with a detection limit of 0.27 μM at the signal-to-noise ratio of 3, and the apparent Michaelis–Menten constant was 0.067 mM. Their work showed that loading metal nanoparticles can further significantly improve the sensitivity and selectivity of the electrochemical sensors, presenting a superior detection performance to the pure approach. Based on the easy modification of CDs, the rich carboxyl or amino functional groups on their surface are conducive to the grafting of small organic molecules for the detection of special chemicals. For example, Zhang et al. [71] selected nitrogen-doped carbon dots (N-CDs) containing amino groups as the raw electrode due to their good conductivity and redox performance. In addition, a Ferrocene (Fc) and β-cyclodextrin (β-CD) host–guest complex was introduced to improve the solubility, electrical stability, and bioavailability. Accordingly, based on the CV and DPV techniques in phosphate buffer solution, this ternary detection system, Fc CD/N-CDs, was found to be a suitable choice for the detection of uric acid via the establishment of a new method. Shao et al. [72] used CDs to interact with *N*-(2-aminoethyl)-*N*,*N′*,*N′*-tris(pyridine-2-yl-methyl)ethane-1,2-diamine (TPEA) for the detection of $Cu^{2+}$. Benefiting from the good conductivity of CDs and the strong complexing ability of TPEA with $Cu^{2+}$, the detection of $Cu^{2+}$ content in the

mouse brain was as good as the result of inductively coupled plasma-atomic emission spectrometry (ICP-AES), indicating that this method has good feasibility and application prospects (Figure 3e). Li and coworkers [73] successfully synthesized the CDs/octahedral $Cu_2O$ nanocomposites, which exhibited good linear response, low detection limit, high selectivity, and wide detection range toward the electrocatalytic oxidation of glucose and the electrocatalytic reduction of $H_2O_2$. Yang et al. [74] prepared an ultra-sensitive sensor of Au NP/CD nanocomposites protected by Fc derivatives and graphene, which can be used for simultaneous detection of vitamin C, DA, UA, and paracetamol.

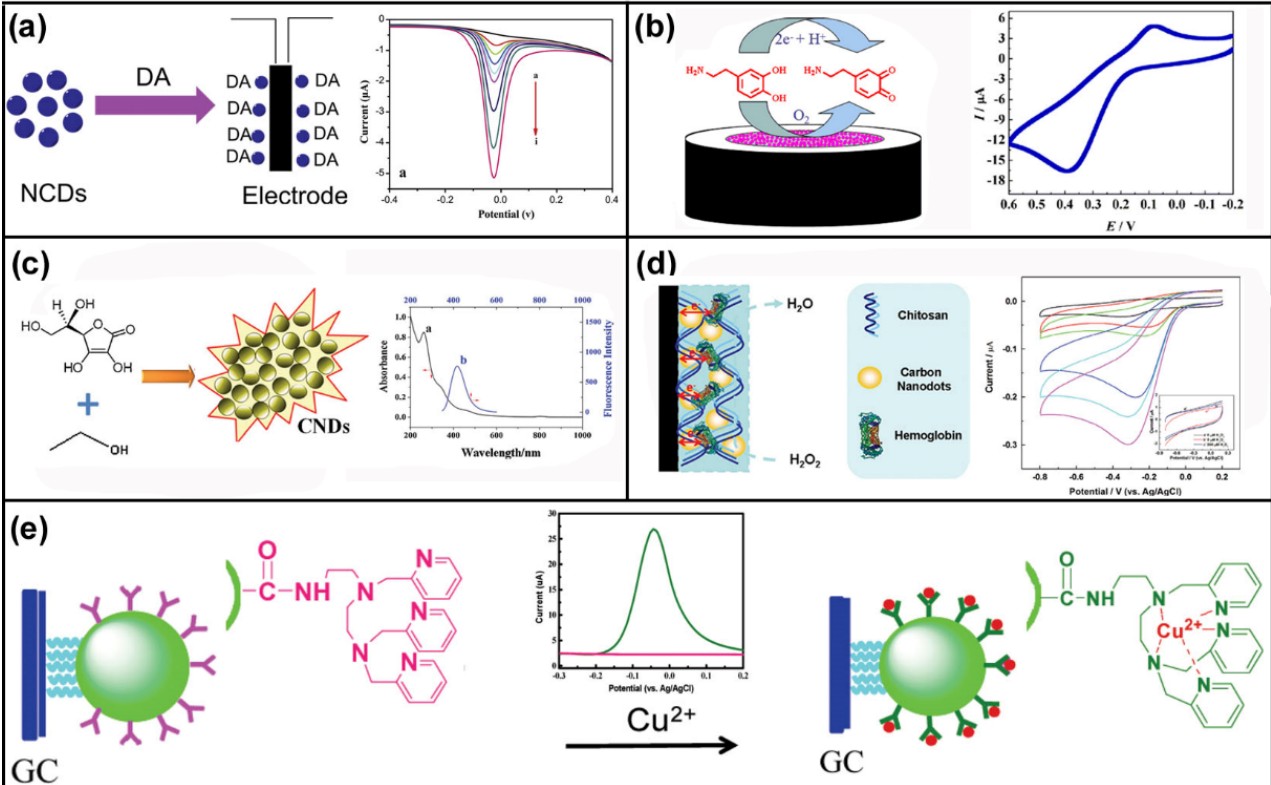

**Figure 3.** CD-based electrochemical sensors for food and alcoholic beverage safety. (**a**) Schematic illustration of N-doped CD synthesis and DA detection. Reprinted with permission from [67]. Copyright Elsevier, 2015. (**b**) Schematic illustration of the strategy for DA detection. Reprinted with permission from [68]. Copyright the Royal Society of Chemistry, 2013. (**c**) Hydrothermal approach to CNDs and its TEM image, UV-vis absorption, and PL spectra. Reprinted with permission from [69]. Copyright Elsevier, 2012. (**d**) Schematic illustration of Hb-CNDs-chitosan system and CVs of Hb-CNDs-chitosan/GC electrode in 5 pH 7.4 PBS containing $H_2O_2$. Reprinted with permission from [70]. Copyright Elsevier, 2014. (**e**) A schematic illustration of the electrochemical sensor for $Cu^{2+}$ detection based on CDs-TPEA. Reprinted with permission from [72], Copyright American Chemical Society, 2012.

As summarized in Table 1, CDs have been widely used as electrode materials for the detection of chemical substances in bulk food, but their application to alcoholic beverages still needs to be further studied. Modifications such as doping nitrogen, grafting organic molecules, and loading metal nanoparticles can greatly improve the sensitivity, selectivity, durability, and repeatability of CD-based electrochemical sensors. However, the resultant complex electrode systems greatly hinder the exploration of underlying detection mechanisms. Therefore, in the future, more attention should focus on the development of electrode systems with an accurate structure and single components.

**Table 1.** Typical applications of electrochemical sensors based on CD materials associated with electrochemical techniques for the detection of different target analytes.

| Analytes | Materials | Electrochemical Techniques | Linear Range | Detection Limit | Sample | Ref. |
|---|---|---|---|---|---|---|
| Dopamine | N-doped CDs | DPV | 0.05~8 μM | 1.2 nM | Serum | [67] |
| Dopamine | Au@CDs-CS | DPV and CV | 0.01~100 μM | 0.1 nM | Spiked sample | [68] |
| Triclosan | CNDs-chitosan | CV | 10 nM~1.0 mM | 9.2 nM | Water | [69] |
| $H_2O_2$ | Hb-CNDs-chitosan | CV | 1~118 μM | 0.27 μM | Toothpaste | [70] |
| Uric acid | Fc@β-CD/N-CD | DPV | 5~120 μM | 0.08 μM | Urine | [71] |
| $Cu^{2+}$ ions | CDs-TPEA | DPASV | 1~60 μM | 100 nM | Spiked sample | [72] |
| Glucose | CDs/$Cu_2O$ NPs | CV | 0.02~4.3 mM | 8.4 μM | - | [64] |
| Acetamteinophen | Fc-S-Au/CDs | CV | 0.5~46 μM | 0.1 μM | Urine | [74] |

CDs: carbon dots; CV: cyclic voltammetry; mM: millimolar; μM: micromolar; nM: nanomolar; $H_2O_2$: hydrogen dioxide; CNDs: nitrogen-doped carbon dots; Hb: hemoglobin; Fc: ferrocene; DPV: differential pulse voltammetry; TPEA: *N*-(2-aminoethyl)-*N*,*N*′,*N*′-tris(pyridine-2-yl-methyl)ethane-1,2-diamine; DPASV: differential pulse anodic stripping voltammetry; $Cu_2O$: cuprous oxide, CS: chitosan.

### 3.2. CNTs

CNTs, also known as bucky tubes, are one-dimensional tubular nanomaterials with special structures. CNTs are mainly coaxial round tubes with a single to dozens of layers composed of sp$^2$ hybridized carbon atoms [75,76]. In theory, CNTs can be regarded as hollow tubes rolled by graphene sheets. According to the number of rolled layers of the graphene sheets, CNTs can be subdivided into single-walled carbon nanotubes (SWCNT) formed by a hexagonal grid structure, and multi-walled carbon nanotubes (MWCNT) assembled from several to dozens of concentric cylinders with regular layer spacing [77,78]. Because their unique structure is completely different from that of bulk carbon materials, CNTs can exhibit excellent properties, such as unique electrical, special magnetic, and strong light absorption properties. Because of their excellent electrical conductivity, large specific surface area, good biocompatibility, easily functionalization, and abundant active sites, CNTs are also an advantageous option in the design of electrochemical sensors [79–81]. In the following section, we illustrate the applications of CNTs-based materials in the development of electrochemical sensors for food and beverage safety, as summarized in Table 2.

Wine is a complex drink mixed with hundreds of compounds, among which, phenolic compounds exist at much lower levels of concentration than other organics [82]. To evaluate the antioxidant properties of the wine samples, the gallic acid (GA, as one of the main phenolic components) oxidation process was constructed using 30% (m/m) carbon nanotube-modified electrodes [83]. The linear range of 0.5–15 μM and the detection limit of 0.3 μM were obtained in an optimized experimental condition. Finally, the proposed procedure was successfully used for estimating the determination of total polyphenols in the samples of red and white wines. Compared with traditional analytic methods, electrochemical analytic methods with CNT-modified electrodes have the advantages of high detection limit, small demand for raw materials, and strong selective identification of specific pollutants [84]. Furthermore, Ziyatdinova and co-workers [85] researched the oxidation peaks of phenolic antioxidants existing in red or white dry wine samples based on MWNT/GCE using differential pulse voltammetry in the phosphate buffer solution of pH 4.0. The evaluation of wine antioxidant capacity (AOC) was established by a one-step chronocoulometric method. The results demonstrated that the AOC of white wine was significantly less than that of red wine (386 ± 112 vs. 1224 ± 184, $p < 0.0001$). The presented effective measurement, AOC evaluation, made it possible for quality control at different stages in the winemaking process, which usually cannot be accurately performed due to the limitation of common analytic technology. It should be emphasized that CNT-based composite materials via introducing metal oxides and metal nanoparticles have recently shown great success in food and beverage safety testing. The possible reasons for this were that the added metal components dramatically boosted electrical conductivity, enhanced chemical signals, and promoted chemical interaction with the analyte. As shown in Figure 4a, various phenolic compounds in wine samples were determined by CV, DPV,

and EIS methods based on $Fe_3O_4$/modified carbon paste electrode (MCPE) in a three-electrode system. In their work, the incorporation of $Fe_3O_4$/MCPE electrodes as a sensor showed excellent sensitivity, selectivity, repeatability, reproducibility, stability, and low preparation cost via optimizing scan rates and pH values. The detection limits equating to 2.2–10 µM for sinapic acid, 2.6–10 µM for syringic acid, and 0.8–10 µM for rutin were obtained. [86] Moreover, various types of metal nanoparticles were added onto the CNTs to generate additional electrocatalytic sites and to increase the sensitivity and detection limits of resultant electrodes [87]. A CS nanocomposite with Au-NP-decorated CNTs has been developed as an electrochemical sensor for the detection of catechol. Due to highly electrochemical active AuNPs, the excellent reproducibility and repeatability of catechol detection in the range of 0 to 1 mM were obtained by CV analysis, with a detection limit of 3.7 µM [88]. Ezhil Vilian et al. [89] reported $Pt/MnO_2$/functionalized multi-wall CNT (f-MWCNT) electrode material was synthesized by a simple and facile strategy. As shown in Figure 4b, the as-prepared $Pt/MnO_2$/f-MWCNT, having a larger effective surface area, greater porosity, and more reactive sites than bare MWCNTs, exhibited excellent sensitivity for catechin sensing. Under optimized conditions, a very low detection limit (0.02 µM) and the linear range of 2–950 µM were obtained. The developed electrochemical sensor was expected to be useful for industrial applications because of the electrochemical reactivity, excellent reproducibility, and good long-term stability. Similarly, as shown in Figure 4c, a novel electrochemical sensor based on GOX-NFM/MWCNT composite was developed by electrospinning and coating methods to detect glucose in beer samples. The electrochemical response of the sensor was analyzed by CV and CA, and excellent reproducibility and repeatability to glucose detection in the range of 1–3 mM were found with a detection limit of 20 µM [90].

**Table 2.** Application of electrochemical sensors based on CNT-based materials associated with electrochemical techniques for the detection of different target analytes in foods and beverages.

| Analytes | Materials | Electrochemical Techniques | Linear Range | Detection Limit | Sample | Ref. |
|---|---|---|---|---|---|---|
| Gallic acid | MCPE | CV and DPV | 0.5–15 µM | 0.3 µM | Wines | [84] |
| Phenolic antioxidants | MWCNT | DPV | ND | ND | Wines | [85] |
| Phenolic compounds | $Fe_3O_4$/MCPE | CV and DPV and ESI | 0.22–0.26 µM | 0.08 µM | Wines | [86] |
| Catechol | AuNP-MWCNT | CV | 0–1.0 mM | 3.7 µM | Wines | [88] |
| Catechin | $Pt/MnO_2$/f-MWCNT | SWV | 2–950 µM | 0.02 µM | Wines | [89] |
| Glucose | GOX-NFM/MWCNT | CV and CA | 1–3 mM | 20 µM | Beer | [90] |
| TBHQ | CuO NFs/$NH_2$-CNTs | DPV | 0.01–3.9 µM | 3 nM | Edible oils | [91] |
| Bisphenol A | MWCNTs-βCD/SPCE | CV | 125 nM–2 µM | 13.76 nM | Water | [92] |
| Methyl parathion | MWCNT/zirconia | CV | 19.9–176.8 µM | 9 nM | Ethanolic soybean | [93] |
| Semicarbazide | MIP/SWNTs-COOH/CS | CV and PDV and ESI | 0.04–0.6 ng mL$^{-1}$ | 0.025 ng mL$^{-1}$ | Sheep casings | [94] |

MCPE: modified carbon paste electrode; CV: cyclic voltammetry; CA: chronoamperometry; DPV: differential pulse voltammetry; SWV: square wave voltammetry; ESI: electrochemical impedance spectroscopy; mM: millimolar; µM: micromolar; nM: nanomolar; ng: nanogram; MWCNT: multi-walled carbon nanotube; $Fe_3O_4$: ferroferric oxide; AuNPs: gold nanoparticles; Pt: platinum; $MnO_2$: manganese dioxide; GOX: glucose oxidase; NFM: nylon nanofibrous membrane; TBHQ: *tert*-butyl hydroquinone; CuO: copper oxide; NFs: nanoflowers; βCD: β-cyclodextrin; SPCE: screen-printed carbon electrode; MIP: molecularly imprinted polymer; SWNTs: single-walled carbon-nanotubes; CS: chitosan.

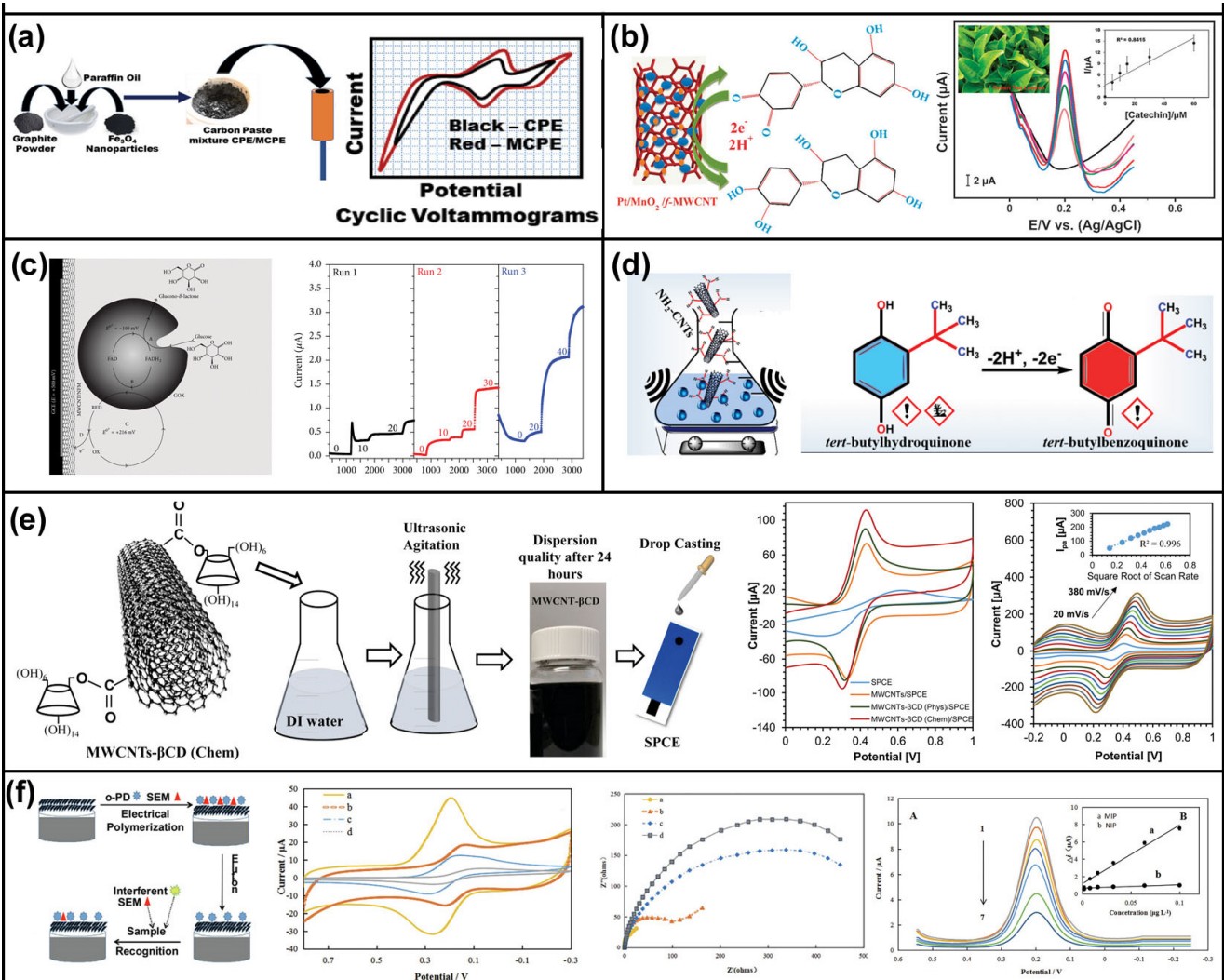

**Figure 4.** CNT-based electrochemical sensors for food and alcoholic beverage safety. (**a**) Schematic diagram of the phenolic compounds' electrochemical sensor preparation. Reprinted with permission from [86]. Copyright MDPI, 2021. (**b**) SWV experiments for various concentrations of 0, 5, 10, 15, 30, and 60 mM of green tea leaves extract diluted with catechin concentration. Reprinted with permission from [89]. Copyright The Royal Society of Chemistry, 2015. (**c**) Sensing mechanism. OX and RED are the oxidized and reduced mediator forms, respectively. Reprinted with permission from [90]. Copyright Hindawi, 2016. (**d**) Schematic illustration of TBHQ detection. Reprinted with permission from [91]. Copyright Elsevier, 2021. (**e**) CV curves of different electrodes and different scan rates in an aqueous electrolyte. Reprinted with permission from [92]. Copyright Elsevier, 2020. (**f**) The schematic diagram of the fabrication procedure of the sensor. Reprinted with permission from [94]. Copyright Elsevier, 2020.

Furthermore, Balram et al. [91] reported a novel CuO NF/NH$_2$-CNT composite synthesized using the hydrothermal and ultrasound-assisted method. As shown in Figure 4d, this composite as a working electrode material was used for detecting cytotoxic food preservative *tert*-butyl hydroquinone (TBHQ). In this work, 3D CuO nanoflowers (NFs) with exceptional chemical stability, non-toxicity, and electron correlation effects could improve the reproducibility, testing repeatability, and stability of electrochemical sensors. An excellent recovery efficiency in the range of 95.90–104.87% and a maximum relative standard deviation (RSD) of 2.71% were obtained using Oils analysis. Recently, Bisphenol A (BPA), one of the most extensively used plasticizers, was detected based on the reversibility and irreversibility of electrochemical redox reactions. Younus Alia and co-workers [92]

developed a simple, low-cost electrochemical sensor (Figure 4e) for detecting very low concentrations of BPA in water using a WCNTs-βCD decorated screen-printed carbon electrode (SPCE). The electrochemical response of the sensor for detecting BPA showed a completely irreversible process including two electrons and two protons. The sensor showed a two-step linear response from 125 nM to 2 μM and 2 to 30 μM, with a detection limit of 13.76 nM. Moreover, a mesoporous MWCNT/zirconia composite was synthesized by the sol-gel method to detect methyl parathion (MP) in food samples. In this work, the electrochemical responses for MP were obtained by DPV with a detection limit of 9 nM and in a linear interval of 19.9–176.8 μM. Specifically, the developed sensor was applied in ethanolic soybean extract and showed satisfactory recovery and repeatability [93]. In addition to modifying CNTs with metal or metal oxide nanoparticles, directly installing active molecules on CNTs was also a very effective method for the rapid detection of contaminants in food. Recently, as shown in Figure 4f, Yu et al. [94] reported that molecularly imprinted polymer (MIP)/carboxylated single-walled carbon-nanotubes/chitosan (MIP/SWNTs-COOH/CS) was prepared as an electrochemical sensor using MIP as the recognition element. Three different electrochemical techniques, including CV, DPV, and ESI, were applied for the detection of semicarbazide (SEM) in four different real samples. The linear range of 0.04-0.6 ng mL$^{-1}$ and the low detection limit of 0.025 ng mL$^{-1}$ were obtained at an optimized experimental condition.

Based on the good conductivity and strong recyclability of carbon nanotubes, we first reviewed the detection and monitoring of chemicals in foods and alcoholic beverages using CNT-based materials as sensor electrodes. Furthermore, a variety of electrochemical analysis methods used for qualitatively and quantitatively detecting chemical substances in foods were also introduced. In addition, compared with CDs, CNTs can be directly used to detect the antioxidant and bioactive substances in alcoholic beverages due to the multiple active sites exposed on CNTs surface. Although a large number of advances have been achieved, more strategies aiming to construct a specific and effective sensor electrode remain to be further studied.

### 3.3. Graphene

Graphene is a 2D honeycomb lattice structure composed of a single layer of sp$^2$ hybridized carbon atoms. Graphene-based materials have been widely examined by researchers in experiments and theoretical studies due to, among other factors, their high chemical stability, thermal conductivity, and electron mobility [95,96]. As far as we know, graphene has shown great application potential in the fields of electronics, optics, magnetism, biomedicine, catalysis, and sensors [97,98]. In particular, graphene is very suitable for the development of sensing materials in electrochemical sensors by taking advantage of its large specific surface area and high electron mobility. In the following section, we illustrate the applications of graphene-based materials in the development of electrochemical sensors for food and beverage safety, as summarized in Table 3.

**Table 3.** Application of electrochemical sensors based on graphene-based materials associated with electrochemical techniques for the detection of different target analytes in foods and beverages.

| Analytes | Materials | Electrochemical Techniques | Linear Range | Detection Limit | Sample | Ref. |
|---|---|---|---|---|---|---|
| Atropine | Graphene-PLA | SWV | 5–60 μM | 1 μM | Wines | [99] |
| Amino acids | Nanographene | CA | $LR_h$, $LR_l$, $LR_o$, $LR_t$ | $DL_h$, $DL_l$, $DL_o$, $DL_t$ | Wines | [100] |
| Gallic acid | CS–fFe$_2$O$_3$–ERGO | DPV | 1–100 μM | 0.15 μM | Wines | [101] |
| Aflatoxin B1 | FGO | CV and PDV and ESI | 0.05–6.0 ng mL$^{-1}$ | 0.05 ng mL$^{-1}$ | Wines | [102] |
| Aflatoxin B1 | RGO/MoS$_2$/PANI@Au/Cs | DPV | 0.01–1.0 fg mL$^{-1}$ | 0.002 fg mL$^{-1}$ | Wines | [103] |
| Saccharomyces cerevisiae | PA-GO/SPE | CA | 10–107 CFU mL$^{-1}$ | ND | White wine | [104] |
| *trans*-Resveratrol | LPG | CV and DPV | 0.2–50 μM | 0.16 μM | Red wine | [105] |
| *trans*-Resveratrol | Gr-MoS$_2$ | DPV | 1.0–200 μM | 0.45 μM | Red wine | [106] |
| Caffeic acid | SnO$_2$-RGO | DPV | 0.15–25 μM | 80 nM | Red wine | [107] |
| Fe$^{3+}$ | po-Gr-NR | CV and DPV | 37.5 nM–21.53 mM | 18.7 nM | Red wine | [108] |
| Sunset yellow and Tartrazine | GN/TiO$_2$ | CV and SWV | $LR_{sy}$, $LR_{tt}$ | $DL_{sy}$, $DL_{tt}$ | Foods | [109] |
| Bisphenol A | GNPs/GR | CV | 0.01 μM–10 μM | 5 nM | Milk | [110] |
| TRA | GCE\|Gr-Au/MIPs | CV | 0.01–10 μM | 0.0044 μM | Foods and medicines | [111] |
| Vitamin C | Au NPs/PCA-RGO | CV | 50–500 μM | 17 μM | Foods | [112] |
| Aflatoxin B1 | AuNPs/rGO/ITO | CV | Nr | 6.9 pg mL$^{-1}$ | Foods | [113] |

PLA: polylactic; CV: cyclic voltammetry; CA: chronoamperometry; DPV: differential pulse voltammetry; SWV: square wave voltammetry; ESI: electrochemical impedance spectroscopy; mM: millimolar; μM: micromolar; nM: nanomolar; pg: picogram; fg: femtogram; CS: chitosan; Fe$_3$O$_4$: ferroferric oxide; ERGO: electrochemically reduced graphene oxide; FGO: functional graphene oxide; RGO: reduced graphene oxide; MoS$_2$: molybdenum disulfide; PANI: polyaniline; AuNPs: gold nanoparticles; PA-GO: propionic acid-functionalized graphene oxide; LPG: laser-induced porous graphene; po-Gr-NR: NR-treated partially oxidized graphene; GCE: glassy carbon electrode; MIP: molecularly imprinted polymer; PCA: 1-pyrene carboxylic acid; $LR_h$:0.1 nM–0.1 μM; $LR_l$: 0.1–100 μM; $LR_o$: 0.01–100 μM; $LR_t$: 0.1 nM–10 nM; $DL_h$: 0.1 nM; $DL_l$: 0.1 μM; $DL_o$: 10 nM; $DL_t$: 0.1 nM; $LR_{sy}$: 0.02–2.05 μM; $LR_{tt}$: 0.02–1.18 μM; $DL_{sy}$: 6 nM; $DL_{tt}$: 8 nM.

Due to the 2D planar structure of graphene, the fabricated detection system can fully contact the detected substances, which is conducive to accelerating mass transfer and therefore realizing rapid detection. In addition, the detection requirements of different pollutants can be rationally met via grafting organic functional groups or compounding different metal nanomaterials. Recently, Cioates Negut et al. [100] reported that oleamide-derivative decorated graphene can be directly coated on carbon electrodes and used to detect amino acids, such as L-histidine, L-tyrosine, L-ornithine, and L-lysine in wine (Figure 5a). In this work, the proposed sensors exhibited excellent sensitivity, reliability, and reproducibility for the quality determination of wines. Furthermore, an amperometric immunosensor for Saccharomyces cerevisiae was constructed using functionalizing and coupling approaches, as shown in Figure 5b. The immunosensor allowed the amperometric detection of the yeast in buffer solution and white wine samples in the range of 10–107 CFU mL$^{-1}$. Because this propionic acid-functionalized graphene oxide could provide a large number of adsorption and sensing sites, the immunosensor successfully detected Saccharomyces cerevisiae at the refermentation stage, exhibiting a low detection limit and high selectivity, and good reproducibility and storage stability [104]. The functionalized graphene is not merely able to serve as a platform for aptamer sequencing, but also as the signal-enlarging platform. Goud et al. [102] reported an electrochemical apta-sensor based on hexamethylenediamine-functionalized GO as the signal-enlarging platform via carbodi-imide amide-bonding. Aflatoxin B1 (AFB1) analyte molecule detection was accomplished in phosphate buffer saline (PBS) solution using methylene blue as a signaling fragment and FGO as the signal-enlarging platform. Under optimized conditions, a very low detection limit was obtained with the linear range of 0.05–6.0 ng mL$^{-1}$. Heteroatom doped graphene is one of the significant strategies for graphene properties' regulation. Nitrogen, phosphorus, oxygen, sulfur, and other atom-doped graphene have been extensively studied in the

electrochemical catalysis field [114]. Identification and detection of iron (III) (ferriciron, $Fe^{3+}$) are challenging in the complex system using traditional detection techniques. As shown in Figure 5c, the $Fe^{3+}$ electrochemical sensor was constructed by depositing the partially oxidized graphene sheets (po-Gr) on a glassy carbon electrode. The linear response range and the limit of detection of $Fe^{3+}$ were obtained from 37.5 nM to 21.53 mM and 18.7 nM in the presence of other valence ions, such as $Fe^{2+}$, $Cu^{2+}$, $Pb^{2+}$, $Hg^{2+}$, $Mn^{2+}$, $Ni^{2+}$, $Zn^{2+}$, $Co^{2+}$, and $Cd^{2+}$, respectively. Interestingly, in this work, partially oxidized graphene sheets provided active sites for selectively capturing $Fe^{3+}$, allowing it to be detected in low concentrations in complex red wine samples [108].

Metal oxides are stable heterogeneous catalysts with abundantly exposed active sites, and have been extensively studied in photocatalytic, electrochemical, and traditional thermal catalytic reactions. As shown in Figure 5d, the CS-fishbone-shaped $Fe_2O_3$ (f$Fe_2O_3$)-electrochemically reduced graphene oxide (ERGO), having a large surface area, excellent electronic conductivity, and high stability, was modified on the glassy carbon electrode for detecting GA to estimate the antioxidant capacity of wines. Under the optimal conditions, the good linear range and the low limit of detection of GA were obtained from 1 to 100 µM and 0.15 µM, respectively. In this work, ERGO and f$Fe_2O_3$ hybrids have a significant synergistic amplification effect on enhancing the electrochemical performance of the electrode. Moreover, the good structural stability of the reported CS-f$Fe_2O_3$-ERGO can avoid contamination of the detection system [101]. Similarly, Zhang et al. [107] reported an electrochemical sensor that was fabricated using $SnO_2$ decorated graphene ($SnO_2$-RGO) composite, which exhibited excellent selectivity, reproducibility, and stability for caffeic acid (CA) detection in commercial red wine samples.

Two-dimensional disulfide nanosheets, and particularly molybdenum disulfide ($MoS_2$), have aroused tremendous research interest in electrochemical aptasensor applications under complicated conditions [115,116]. However, the facile aggregation and low electronic conductivities of $MoS_2$ decrease the number of electroactive sites on the electrode surface, thereby restricting electron transfer and related electrochemical kinetics [117,118]. A highly sensitive electrochemical sensor based on 2D nanocomposite for *trans*-resveratrol (TRA) was fabricated using graphene-molybdenum disulfide (Gr-$MoS_2$), as shown in Figure 5e. Under optimized conditions, the prepared sensor showed a linear response in TRA concentration from 1.0 to 200 µM with a limit of detection of 0.45 µM due to the synergistic effect between Gr and $MoS_2$ [105]. Furthermore, Wang et al. [103] reported reduced graphene oxide/molybdenum disulfide/polyaniline@gold nanoparticles/chitosan (RGO/$MoS_2$/PANI@AuNPs/Cs), among which Cs acted as an aptasensor and AuNPs were dedicated to signal amplification, was constructed using facile hydrothermal and coating methods. The Au-S bonds formed on the surface of $MoS_2$ facilitate effective installation of the aptamer. As-developed RGO/$MoS_2$/PANI@AuNPs/Apt exhibited a wide linear range from 0.01 fg mL$^{-1}$ to 1.0 fg mL$^{-1}$ and a remarkably low detection limit of 0.002 fg mL$^{-1}$ due to the synergistic effects among distinct components.

In addition to coating catalysts on the surface of carbon electrodes, fabricating self-supported electrodes can effectively increase the contact between the detected substance and the sensor. Based on 3D printing technology, graphene-polylactic acid (graphene-PLA) electrodes were fabricated for detecting atropine in contaminated beverage samples, such as white wine, vodka, whisky, and energy drink. A linear concentration range between 5 and 60 µM associated with a detection limit of 1 µM was obtained using a SWV determination protocol [109]. Moreover, as shown in Figure 5f, a novel flexible electrochemical sensor using a direct laser-induced graphene (LIG) technique that transforms the commercial Kapton tape into 3D porous graphene was developed for sensitive detection of *trans*-tesveratrol (TRA) molecules in red wines. The prepared electrochemical sensor with excellent repeatability, stability, reproducibility, and reliability guaranteed an excellent linear response within the TRA concentration range from 0.2 to 50 µM and a low limit of detection of 0.16 µM. Furthermore, the developed sensor can be applied for the evaluation of TRA levels in red wines and grape skins with a satisfactory result [106].

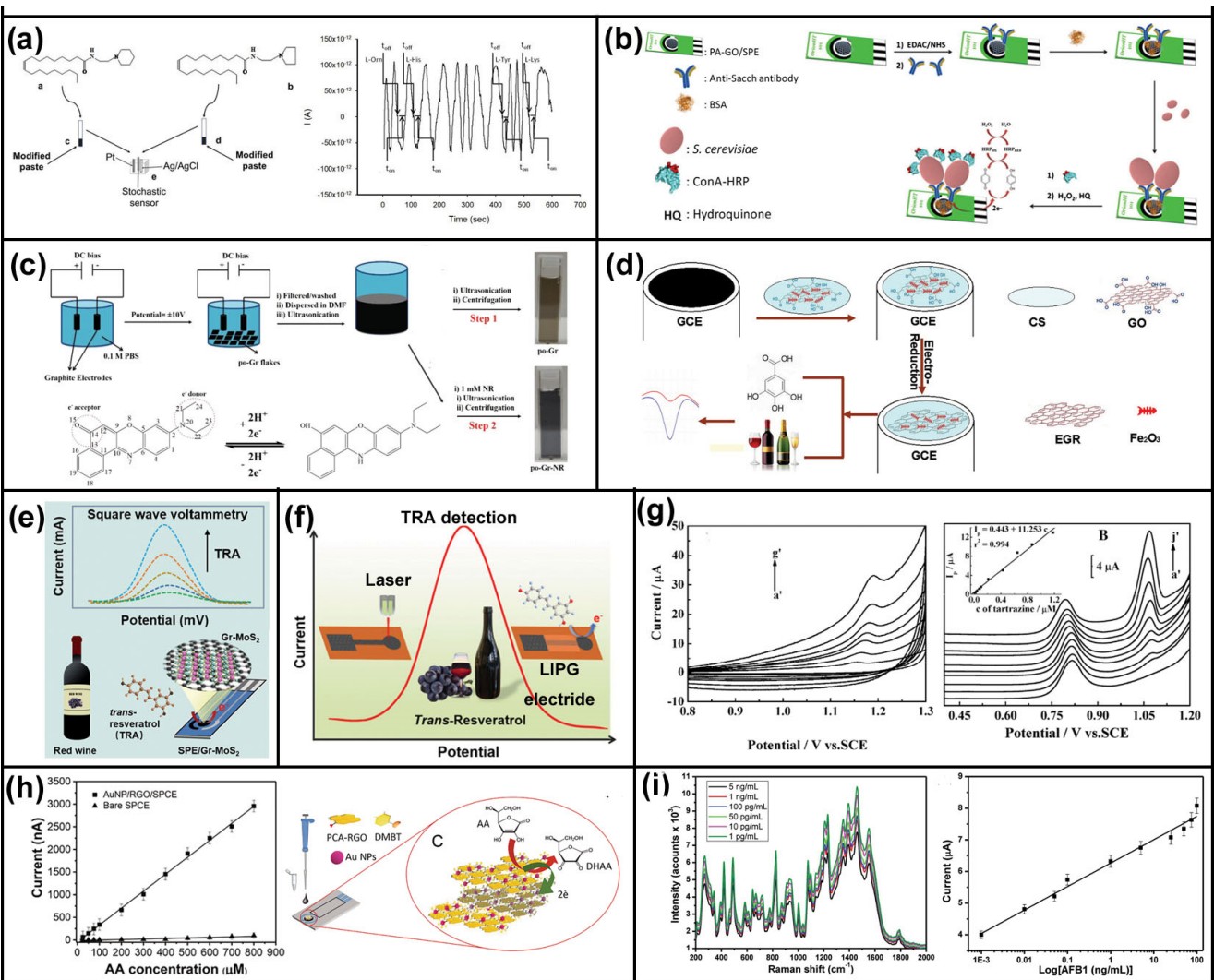

**Figure 5.** Graphene-based electrochemical sensors for food and alcoholic beverage safety. (**a**) The design of the electrochemical platform. Reprinted with permission from [100]. Copyright Wiley, 2019. (**b**) Schematic display of the steps involved in the preparation and performance of the immunosensor for Sacch. Reprinted with permission from [104]. Copyright Springer, 2018. (**c**) Electrochemical synthesis of po-Gr and treatment of NR with po-Gr. Reprinted with permission from [108]. Copyright Elsevier, 2017. (**d**) Illustration for the fabrication of CS–fFe$_2$O$_3$–ERGO/GCE and its application for the detection of GA in wines. Reprinted with permission from [101]. Copyright Elsevier, 2015. (**e**) Two-dimensional nanocomposite-based electrochemical sensor for rapid determination of trans-resveratrol. Reprinted with permission from [106]. Copyright Elsevier, 2020. (**f**) Schematic illustration of the preparation of LIPG-based electrochemical sensor for the detection of TRA in red wines and grape skins. Reprinted with permission from [105]. Copyright Elsevier, 2020. (**g**) CV curves and the possible electrochemical reaction mechanism of sunset yellow and tartrazine. Reprinted with permission from [109]. Copyright Elsevier, 2013. (**h**) Sketch of the AuNP/RGO/SPCE-modified nanostructured platform and its application for AA determination in milk. Reprinted with permission from [112]. Copyright Elsevier, 2020. (**i**) Raman spectrum and CV curves for different concentrations of AFB1. Reproduced with permission from [113]. Copyright PLOS, 2019.

Electrochemical sensors based on graphene composite materials have been widely studied in daily food safety and drug monitoring. In order to maximize the detection efficiency and accuracy, porous structures such as Co$_3$O$_4$, TiO$_2$, and MnO$_2$ were loaded onto the surface of graphene to exhibit greater analytic performance in food and alcohol

safety [119]. For instance, a composite of graphene and mesoporous $TiO_2$, as a novel voltammetric sensor, was fabricated by a facile one-pot hydrothermal method. The developed sensor exhibited well-defined and separated SWV peaks (i.e., 272 mV) for detecting sunset yellow FCF and tartrazine in several food sample extracts using CV and SWV techniques, as shown in Figure 5g [109]. With the aim of enhancing the sensitivity of the electrochemical apta-sensors, Zhou et al. [110] reported gold nanoparticles dotted graphene (GNPs/GR) as an efficient aptamer used for bisphenol A (BPA) detection in milk products. The fabricated electrochemical sensor exhibited a wide range of concentrations and a low limit of BPA detection by analyzing the current change of ferricyanide. Similarly, Yang et al. [111] reported the composite of layer-by-layer films of graphene (Gr)-gold nanoparticles (Au) and molecularly imprinted polymers (MIPs) as an electrochemical sensor used for efficiently detecting trans-resveratrol by taking advantage of their synergistic effects. In addition to the detection of chemical contaminants in foods, it is also important to monitor the quality of nutrients in food. Au nanoparticles decorated reduced graphene oxide flakes as an efficient detector were developed to monitor the quality control and quantitative assessment of vitamin C in infant food by studying L-ascorbic acid (AA) oxidation. As shown in Figure 5h, the oxidation of AA to dehydroascorbic acid showed a good linear relationship in the range of 50–500 μM and a low detection limit of 17 μM [112]. Finally, combining electrochemical sensing technology with other technologies, such as Raman spectroscopy, FT-IR spectroscopy, and mass spectrometry, can further improve the detection accuracy and sensitivity. A new Aflatoxin B1 sensor was developed based on Au nanostructures/graphene nanosheets-modified indium tin oxide (ITO) substrate, which could enhance the Raman effect and the electrochemical conductivity (Figure 5i). The presented sensor was a simple, easy, and sensitive sensor for monitoring the low concentrations of AFB1 with a detection limit of about 6.9 pg mL$^{-1}$. It also allowed the determination of AFB1 in spiked food samples [113].

Graphene is an excellent candidate in the design of electrochemical sensors due to its excellent electrical conductivity, large specific surface area, good biocompatibility, easy functionalization, and abundant active sites. Benefiting from the development of functionalization strategies, highly efficient electrochemical sensors can be constructed by doping heteroatoms on graphene. The heteroatom-doped graphene with an accurate structure helps to explore the sensing mechanism at the molecular level. In addition, metal nanoparticles or metal oxide particles are usually loaded on the surface of graphene to improve the sensitivity of electrochemical sensors. However, the added metal salts will greatly increase the cost and even deteriorate the stability of sensors. In future research, it is hoped that researchers will directly use graphene as a 3D self-supported electrode in electrochemical sensors based on graphene's robust durability, which will provide technical support for further practical industrial applications.

## 4. Conclusions and Perspectives

In summary, electrochemical sensors constructed by carbon-based composites can acquire a wide linear response range and low detection limit in the detection of food and alcoholic beverage contaminants. CV, LSV, SWV, and DPV are the most widely used electrochemical sensing methods because they can discern the possible intermediates with an extremely low detection limit, decode the nature of coupled chemical reactions, explore the weak adsorption phenomenon, study the mechanism of complex electrode reactions, and suppress the background current, respectively. Due to the good electrical conductivity, wide potential window, and easy chemical modifiability, carbon material (i.e., CDs, CNTs, and graphene)-based composite electrodes have been built with the introduction of metal nanoparticles, polymers, organic acids, doped heteroatoms, metal-oxide nanoparticles, and 2D sulfides to offer high sensitivity, unique selectivity, good durability, and repeatability. The presented electrochemical sensors have been widely used for detecting and monitoring chemical contaminants including dopamine, acetaminophen, $H_2O_2$, GA, methyl parathion, $Cu^{2+}$ and $Fe^{3+}$, Bisphenol A, sunset yellow FCF, Aflatoxin B1, and amino acids in foods and

alcoholic beverages. Furthermore, electrochemical sensors have been practically applied for the detection of chemical contaminants in white wine, red wine, beer, edible oils, water, milk, and medicines.

Although most of the reported sensors exhibited potential as portable instruments, most of them have been assessed only in the laboratory. Thus, developing novel construction methods and revealing the sensing mechanism underlying the molecular precision are the core contents that must be studied continuously in order to realize applications in the industrial model for food and beverage safety in the future. Due to the good electrical conductivity and chemical modifiability of carbon materials, it is hoped that an electrochemical sensor can be constructed based on single-atom functionalized carbon materials, which can further reduce the cost and increase the detection sensitivity. Furthermore, using advanced in situ characterization techniques, such as Fourier-transform infrared spectroscopy (FT-IR), Raman spectroscopy, and Extended X-ray Absorption Fine Structure (EXAFS) to reveal the sensing mechanism, can provide the theoretical support for the rational design and construction of efficient sensing electrodes. Finally, to realize the online detection under complex working conditions available for industrial applications, the proposed integration of designed electrochemical sensors within a smart device, such as a phone, iPad, and computer, is becoming a popular research topic in this field.

**Author Contributions:** Z.Y., J.G. and X.Z. all contributed to the collection of data and preparation of the paper. All authors have read and agreed to the published version of the manuscript.

**Funding:** This work was supported by the Wenzhou Key Laboratory of Biomaterials and Engineering (Grant No: WIUCASSWCL21005) and the Science and Technology Plans of Tianjin (21ZYJDJC00050).

**Institutional Review Board Statement:** Not applicable.

**Informed Consent Statement:** Not applicable.

**Data Availability Statement:** All the data presented in this manuscript were derived from the indicated articles, which are published in the literature and listed in the Reference section.

**Conflicts of Interest:** The author declares no conflict of interest.

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
