# Peer review of "Functionalized Carbon-Based Electrochemical Sensors for Food and Alcoholic Beverage Safety"

_applsci, doi:10.3390/app12189082_

Round 1

Reviewer 1 Report

Comments for authors

Ref. no. applsci-1541973

Title: Functionalized Carbon-based Electrochemical Sensors for Food and Alcoholic Beverage Safety

Overview

In this review, the authors discussed the relationship between the structure of carbonaceous materials and their application in electrochemical sensors for food and alcoholic beverage safety. The authors briefly introduce the principles of the electrochemical sensors, which I believe could be more deeply introduced and discussed. Posteriorly, they present carbonaceous materials, such as carbon dots, carbon nanotubes, graphene, and related examples of applications. Especially, I recommend rewriting all section 3 since it contains a lot of disconnected information which can potentially confuse the reader. The section practically describes shortly the main results of the literature rather than discussing the main challenges, methods and improvements in the field. Although I believe the review is interesting, several points need to be addressed in this current version, as below.

Suggestions and comments

  1. Line 96. Please, revise the sentence “the electrochemical sensors convert the …upon applying electricity into recognizable electrical signals”. The term electricity is too general and it is not accurate. As you have discussed in the text, electrochemical sensors are based on different methods, such as electrochemical impedance spectroscopy (EIS) and voltammetry, in which different signals are used for target detection. For example, in both voltammetry and EIS, the applied signal is potential, and the measured signal is current. Please, revise accordingly.
  2. Please, revise Figure 1c. The CV curve is more representative of a supercapacitor than a sensor. Also, it could be interesting for the readers the authors include the input and output signals for each type of electrochemical sensor.
  3. Please, provide all figures with a better resolution.
  4. Figure 2a. Could you please explain what are the red dots?
  5. Line 144, Section 3. Please, introduce briefly the classification of carbon materials according to their dimensionality.
  6. Table 1. Please, include the type of sample used to detect/quantify the analytes.
  7. Figures 3, 4, and 5 Please improve the text readability.
  8. Tables 2 and 3. Please improve the text readability.
  9. Please, consider rewriting section 3 as discussed above.

Author Response

We gratefully appreciate the reviewers’ comprehensive reports and constructive suggestions. Following all of the reviewers’ constructive suggestions, we have thoroughly revised our manuscript by taking the entire comments into consideration. Please check the revised manuscript in the attachment.

Author Response

We gratefully appreciate the reviewers’ comprehensive suggestions. Following all of the reviewers’ constructive suggestions, we have thoroughly revised our manuscript by taking the entire comments into consideration, please check the revised manuscript in the attachment.  

Reviewer 3 Report

In this manuscript, publications on the electrochemical sensors employing graphene-like electrodes for food and drinks safety analysis were reviewed.
The review is comprehensive, reliable and interesting. I have found only a couple of typos, which will be easily spotted during final preparation of the manuscript for publication.

Author Response

(The authors gave the same response as above.)

Reviewer 4 Report

The manuscript "Functionalized Carbon-based Electrochemical Sensors for Food and Alcoholic Beverage Safety" by Zhongjie Yang et al. is interesting review that might be useful for future readers. However, major revision should be performed before publication. 
The author must improve the keywords.
There are parts of Figures 2 and 3 that are not clear, please improve or simplify the figure. Figures 2, 3, 4, and 5 cannot be included in this way in the manuscript, they are unclear, and each figure contains 6 to 9 illustrations.
Most of the references are old, there are a few recent references but they are not enough. Because this topic has been covered extensively in recent years, the research published between 2019 and 2021 is many times that of previous years. Therefore, the authors should re-select the appropriate recent references.
It is also necessary to expand the presentation and discussion of previous studies. Frankly, the manuscript in its current form does not make a great addition to the subject.

Author Response

(The authors gave the same response as above.)

Round 2

Reviewer 1 Report

The authors have addressed some of my comments and the manuscript has improved. However, I recommend a few minor modifications (e.g. figures formatting) and a major one. Please, see below:

Line 148. Please, include the definition of the materials according to their dimensionality (for example, please see https://doi.org/10.1038/nnano.2009.24).
Please, consider improving the text readability of Figures 2, 3, 4, and 5. For example, the text sizes are too small in some of these figures and hence difficult to read.
In my opinion, section 3 still needs revision. The authors have done some improvements but the section only reports disconnected examples and no deep discussion about the topic, which could be beneficial for the readers whilst improving the paper impact. For example, and not limited to, consider the provided example of DA quantification. For interested readers, it could be important the authors provide more details about the method for analyte quantification, discuss the electrochemical signal used for quantification, and compare with other methods to clarify what are the improvements achieved by using carbonaceous materials. All information is relevant for readers who intend to use such materials. Otherwise, the text is just a description of a few examples rather than a comprehensive discussion about the topic. Please, consider a deep revision in section 3.

Author Response

Reply to the reviewers' comments:

For Reviewer:

General comment: The authors have addressed some of my comments and the manuscript has improved. However, I recommend a few minor modifications (e.g. figures formatting) and a major one. Please, see below:

Our response: We are truly grateful for the reviewer’s encouraging comments and constructive suggestions, which help us to greatly improve the quality of this manuscript.

Comments: Line 148. Please, include the definition of the materials according to their dimensionality (for example, please see https://doi.org/10.1038/nnano.2009.24).Please, consider improving the text readability of Figures 2, 3, 4, and 5. For example, the text sizes are too small in some of these figures and hence difficult to read. In my opinion, section 3 still needs revision. The authors have done some improvements but the section only reports disconnected examples and no deep discussion about the topic, which could be beneficial for the readers whilst improving the paper impact. For example, and not limited to, consider the provided example of DA quantification. For interested readers, it could be important the authors provide more details about the method for analyte quantification, discuss the electrochemical signal used for quantification, and compare with other methods to clarify what are the improvements achieved by using carbonaceous materials. All information is relevant for readers who intend to use such materials. Otherwise, the text is just a description of a few examples rather than a comprehensive discussion about the topic. Please, consider a deep revision in section 3.

Our response: We are truly grateful for the reviewer’s encouraging comments and constructive suggestions, which help us to greatly improve the quality of this manuscript. According to the reviewer's suggestions, in order to facilitate readers' understanding, we refer to the references provided by the reviewer to further define the dimension of the materials. In order to improve the readability of Figure 2, 3, 4 and 5, we have redrawn all figures in the reversed manuscript. Furthermore, we added the relevant descriptions in section 3 to give a comprehensive discussion about the topic.

Our reversion: According to the reviewer’s kind suggestion, we have redrawn all figures 3, 4 and 5 in the revised manuscript. Furthermore, we have added the introduction of carbon based sensors according the dimension of materials, and we have supplemented the relevant descriptions in section 3. Please see the yellow-highlighted sentences in the revised manuscript, respectively.

Reviewer 4 Report

The authors have made the necessary modifications and corrections.

Author Response

For Reviewer:

General comment: The authors have made the necessary modifications and corrections.

Our response: We are truly grateful for the reviewer’s encouraging comments and efforts, which help us to greatly improve the quality of this manuscript.

Round 3

Reviewer 1 Report

The authors addressed my suggestions and I have no more comments.